# Development of Diagnostic Tests Provides Technical Support for the Control of African Swine Fever

**DOI:** 10.3390/vaccines9040343

**Published:** 2021-04-02

**Authors:** Zilong Qiu, Zhaoyao Li, Quanhui Yan, Yuwan Li, Wenjie Xiong, Keke Wu, Xiaowen Li, Shuangqi Fan, Mingqiu Zhao, Hongxing Ding, Jinding Chen

**Affiliations:** 1Department of Microbiology and Immunology, College of Veterinary Medicine, South China Agricultural University, Guangzhou 510642, China; qiuzilong@stu.scau.edu.cn (Z.Q.); lizhaoyao@stu.scau.edu.cn (Z.L.); yanqh@stu.scau.edu.cn (Q.Y.); waner20191028012@stu.scau.edu.cn (Y.L.); wenjiexiong@stu.scau.edu.cn (W.X.); wukeke@stu.scau.edu.cn (K.W.); xiaowenlee@stu.scau.edu.cn (X.L.); shqfan@scau.edu.cn (S.F.); zmingqiu@scau.edu.cn (M.Z.); dinghx@scau.edu.cn (H.D.); 2Guangdong Laboratory for Lingnan Modern Agriculture, Guangzhou 510642, China; 3Key Laboratory of Zoonosis Prevention and Control of Guangdong Province, Guangzhou 510642, China

**Keywords:** African swine fever, ASFV, diagnosis, control, ELISA, epidemic situation

## Abstract

African swine fever is a highly contagious global disease caused by the African swine fever virus. Since African swine fever (ASF) was introduced to Georgia in 2007, it has spread to many Eurasian countries at an extremely fast speed. It has recently spread to China and other major pig-producing countries in southeast Asia, threatening global pork production and food security. As there is no available vaccine at present, prevention and control must be carried out based on early detection and strict biosecurity measures. Early detection should be based on the rapid identification of the disease on the spot, followed by laboratory diagnosis, which is essential for disease control. In this review, we introduced the prevalence, transmission routes, eradication control strategies, and diagnostic methods of ASF. We reviewed the various methods of diagnosing ASF, focusing on their technical characteristics and clinical test results. Finally, we give some prospects for improving the diagnosis strategy in the future.

## 1. Introduction

African swine fever (ASF) is a devastating hemorrhagic fever of swine with mortality rates close to 100%, which is caused by African swine fever virus (ASFV), a large double-stranded DNA virus. It causes major economic losses, threatens food security and restricts pig production in affected countries [1]. Due to the high morbidity and mortality induced by ASF and the lack of effective vaccines, this pathogen poses a serious threat to the global pig industry and national economies. The complex virion composition, life cycle and immune evasion mechanism of ASFV have caused huge troubles in the development of vaccines [2].

The ASFV genome is a linear double-stranded DNA molecule with a length ranging from 170 to 193 kbp between different isolates, containing between 150 and 167 open reading frames (ORFs) [3]. There are three regions of the ASFV genome, wherein the central region is a conserved region, flanked by two variable regions. The difference in genome length and gene number is largely due to the gain or loss of a multigene family (MGF) encoded by the virus. Variations in the number of short tandem repeats within or between genes cause some minor length changes [4,5]. The ends of the genome are covalently cross-linked and exist in two forms, which are inverted and complementary to each other, forming imperfectly base-paired hairpin loops [3]. 

ASFV is the unique member of the family *Asfarviridae*, belonging to nucleocytoplasmic large DNA viruses (NCLDVs), and is currently the only known DNA arbovirus [6,7,8]. The ASFV particle possesses an icosahedral morphology with a diameter of approximately 200 nm and is composed of a complex multilayer structure. From the inside to the outside, it is composed of a central nucleoid containing the genome, a thick protein core shell, an inner lipid membrane, an icosahedral protein capsid, and an outer lipid membrane [9]. The core shell is mainly composed of proteolytic products of polyproteins pp220 (CP1475L) and pp62 (CP530R), accounting for about one-third of the total mass of viral particles [10,11], as well as viral cleavage protease encoded by the gene pS273R [12,13]. The protein icosahedral capsid consists of the main capsid component P72 (pB646L) [14], the minor capsid protein P49 (pB438L) and PE120R, among which PE120R is a protein that mediates viral intracellular transport [13,15]. Interestingly, Liu et al. observed the structure of the ASFV icosahedral capsid under a cryogenic electron microscopy (Cryo-EM) and found that it consisted of the major capsid protein P72 and penton protein, which existed in the form of pseudo-hexameric trimers and pentamers, respectively [16]. Penton protein is considered to be protein H240R. In addition, proteins P17 and M1249L are found below the outer coat shell and play an important role in the construction and stability of capsid framework [17]. The ASFV icosahedral capsid is covered by two layers of envelopes—inner and outer envelopes. The CD2v homologue pEP402R localized at the outermost envelope is an important virulence gene, while P12 (pO61R), P17 (pD117L), P54 (pE183L), and P22 (pKP177R) are situated at the inner envelope with strong immunogenicity [13,18].

Currently, there is no vaccine against ASF. Studies have shown that ASFV encodes a variety of proteins that inhibit type I interferon signaling pathways, such as multigene family proteins MGF360, MGF505/530, DP96R and I329L [19,20,21,22]. The protein A224L encoded by ASFV inhibits apoptosis in the early stage of infection [23]. The ASFV has a variety of genotypes, a large number of encoded proteins, and complex and diverse immune escape mechanisms, which escape the elimination of host immune cells and greatly hinder the development of vaccines. This review introduces the epidemic strains of ASFV in different countries, epidemic history, transmission methods, control eradication strategies and diagnostic methods. It focuses on summarizing the characteristics of different diagnostic methods, providing a reference for the selection of appropriate diagnostic methods for clinical testing, and providing theoretical support for the prevention, control and eradication of ASF. 

## 2. Epidemic Strains and Epidemic History of ASFV

The ASFV genome is enormous and complex. It encodes a large number of proteins, which means extreme antigenic and serogroup diversity. At present, ASFV strains of different virulence have been classified based on the hemoabsorption inhibition assay (HAI), and eight serogroups (serogroups 1–8) have been identified [24]. According to the phylogenetic analysis of the C-terminus end of the B646L encoding protein P72, ASFV can be divided into 24 genotypes (genotypes I–XXIV) to date [25,26]. The African continent is the endemic area with all 24 ASFV genotypes. In western and central Africa, only ASFV genotype I is prevalent in these coastal areas or archipelagos, such as Namibia, Cameroon, Nigeria, Benin, Togo, Ghana, Côte d’Ivoire, Burkina Faso, Senegal, and Cape Verde Islands. It is worth noting that although the Republic of Congo is a coastal country, its popular genotypes are I and IX [27]. In the latest report aimed at investigating the genetic diversity of ASFV strains in symptomatic pigs in South Kivu Province of the Democratic Republic of Congo, the ASFV genotype X was detected for the first time in symptomatic pigs [28]. There are great differences between the ASFV isolates in east and southern Africa, among which 13 ASFV genotypes are prevalent in the east Africa and 14 in the southern Africa. Many countries in east and southern Africa often have more than one ASFV genotype, and some even more. Zambia, for example, has the most prevalent genotypes, and seven genotypes have been identified, followed by six in South Africa and four in Mozambique [25,29,30]. Outside of Africa, only genotype I was found in Europe, South America, and the Caribbean, until ASFV was introduced in Georgia from east Africa in 2007 [31]. After entering Georgia, ASF quickly spread to some countries in Caucasus, eastern Europe, and east Asia. All strains from these countries and Georgia belong to genotype II [32,33,34]. 

ASF was first described in Kenya, Africa, in 1909, causing almost 100% of infected domestic pigs to die [35]. Since then, it has quickly spread to other African countries. In the following decades, ASF has been confined to Africa until it was first discovered in Portugal outside the African continent in 1957. The spread of ASF to Portugal is thought to have taken place through transport of infected meat products on African airline flights or ships, which were fed to pigs. Although this invasion of disease was eradicated in 1958, a further outbreak occurred soon in 1960 in Lisbon (Portugal) [36]. Outbreaks of ASF were reported subsequently in a number of other European countries, including Spain (1960–1995), France (1964, 1967 and 1974), Italy (1967, 1969 and 1993), Andorra (1974), Malta (1978), Belgium (1985) and the Netherlands (1986). Until 1995, the disease had been eradicated in all the European countries mentioned above by hunting wild boars and culling infected pigs, except for Sardinia in Italy [37,38,39]. Despite efforts to eradicate it, ASF has existed in Sardinia in the form of endemic epidemics since 1978. The free-range model was thought to play a key role in the persistence of the ASF [40,41]. The latest eradication plan provides more stringent measures to combat disorderly stocking. From December 2017 to February 2020, 4484 illegal pigs were culled, which greatly reduced the spread of ASF among these animals [42,43]. In 1971, ASF spread across the Atlantic to Cuba (1971, 1980), invading the Caribbean for the first time. It spread in the Caribbean and South America, including the Dominican Republic (1978–1981), Brazil (1978–1986) and Haiti (1979–1982), which were soon under control and eradicated [44].

The first case of ASF in Georgia may be related to the port of Poti on the Black Sea. One possibility is that the contaminated swill or food waste on the ship was eaten by pigs near the port [32]. The disease spread rapidly in the Caucasus, Armenia (2007) and Azerbaijan (2008). ASF spread to the southern part of the Russian Federation through wild boar herds near the Caucasus Mountains in 2007. Subsequently, in 2008, domestic pigs in the Russian Federation were infected and spread rapidly to the western and northern Russia. While the ASF continued to spread, domestic pigs in neighboring Ukraine (2012) and Belarus (2013) were successively infected. In early 2014, ASF spread from the Russian Federation to European Union (EU) countries, especially Lithuania and Poland, through contact with wild boars. ASF kept moving southwest, reaching Moldova and Hungary in 2016 and the Czech Republic and Romania in 2017, respectively [38,45]. From 2018 to December 2020, new outbreaks of ASF have occurred in some European countries including Belgium, Bulgaria, Slovakia, Serbia, Greece, and Germany.

In August 2018, the first outbreak of ASF occurred in China. The strains isolated from China belonged to genotype II, which were similar to the strains currently prevalent in Russia and eastern Europe [33]. From 2018 to December 2020, other Asian countries including Mongolia, the Philippines, Democratic People’s Republic of Korea, Lao People’s Democratic Republic, Myanmar, Timor-Leste, South Korea, Cambodia, Vietnam, Indonesia, Papua New Guinea, and India declared outbreaks of ASF and reported to World Organization for Animal Health (OIE). The distribution and recent status of ASF in Africa (2016–2020), Europe (2007–2020) and Asia (2018–2020) are shown in Figure 1. Table 1 summarizes the ASF affected countries of Europe and Asia from 2007 to December 2020, including the current infection status.

## 3. Transmission

ASFV infects domestic pigs and wild boars, including warthogs (*Phacochoerus africanus*) and bushpigs (*Potamochoerus Larvatus*) in Africa, and wild boars (*Sus Scrofa*) in Eurasia. *Ornithodoros* has been confirmed as the biological carrier and reservoir of ASFV. The ancient sylvatic cycle is an important form of transmission that exists in east and southern Africa. The ASFV is maintained in the sylvatic cycle between warthogs and argasid ticks of the *Ornithodoros moubata* complex [47,48]. Juvenile warthogs dwelling in burrows are infected by *Ornithodoros* soft ticks carrying the virus, and transmission to other soft ticks occurs when the ticks take a blood meal from the infected juvenile warthogs with high levels of viremia. Adult warthogs become virus carriers after being infected with ASFV and will not develop clinical diseases. Adult warthogs play a limited role in the transmission of warthog ticks due to the insufficient ASFV titers in their blood to infect feeding ticks. The horizontal and vertical transmission ability between warthogs is weak, and the transmission occasionally occurs through the shedding of soft ticks carried by warthogs [49]. There are few reports about the sylvatic cycle in west Africa. The spread of ASFV in west Africa mainly occurs through direct contact between domestic pigs or indirect contact between pigs and pork products [37]. The local tick, *Ornithodoros erraticus*, is often found in pig houses in the Iberian Peninsula. This tick spread ASFV by sucking pig blood and maintains the long-term existence of ASFV [38]. However, wild boars play a limited role in ASFV persistence in Spain [50].

The ASFV virion is highly stable in vitro, which provides favorable conditions for pig-to-pig transmission. The virus is easily transmitted via contaminated pork products and mixed feed [51]. Experimental studies have shown that direct contact with infectious domestic pigs is an effective mechanism for the transmission of ASFV. Susceptible animals are directly transmitted by ingesting or inhaling the virus in the debris or secretions present in the environment. When contacted pigs are separated from infected pigs by solid partitions to prevent direct contact between pigs, transmission occurs even later. This shows that the specific farm isolation infrastructure, such as fenced places, can delay but not prevent the spread of the diseases [52]. ASFV has been proven to replicate in *Ornithodoros* species, and it is assumed that these could act as vectors to spread diseases. In contrast, mechanical arthropod vectors, such as mosquitoes and flies, do not support the replication of pathogens, but may ingest pathogens from infected animals during feeding and spread them to other susceptible animals on which they feed [51]. In addition, ingesting contaminated feed or swill feeding may play an important role in the transmission of ASF to domestic pigs. ASF is likely to spread between pigs and wild boars through direct contact with infectious wild boars, infectious free-range pigs, and infected pig carcasses improperly handled by farmers or hunters [53]. This study has investigated the relationship between vaccination route, infectious dose, and virulence level. The authors reported that the nasal route led to a higher incidence of ASF than the oral route when challenged with lower infectious doses, indicating that inhalation was more susceptible to infection than ingestion [54]. There are reports confirming that the infectious virus excreted from infected pigs can spread in short distances through the air, which indicates that the transmission through short-distance aerosols may be an important way for ASF to spread in the farm [55]. However, the limit distance of airborne transmission needs further research to confirm. Soft ticks and wild boars play an important role in the spread of African swine fever, especially in Africa. The potential for the development of new sylvatic or pig–tick cycles, for example, in west Africa, requires further investigation. Understanding the epidemic characteristics and transmission routes of ASF in different regions is essential for the development of successful control and eradication programs in the affected areas.

## 4. Control-Eradication Strategies

### 4.1. Africa

The African continent is an area extremely and severely affected by ASF. The virus circulates in forests in east Africa, involving wild boars, warthogs, and *Ornithodoros* ticks, which presents great challenges to the control and eradication of ASF. The Food and Agriculture Organization (FAO), African Union-Interafrican Bureau for Animal Resources (AU-IBAR), and International Livestock Research Institute (ILR) are jointly preparing an initiative program to control ASF in Africa. The main strategy is to control ASF in affected areas and prevent its spread to noninfected areas. It is also very important to raise awareness among farmers, butchers, and other stakeholders in the pig industry for prevention and control of ASF. For the government, it is necessary to strengthen the role of veterinary services in disease detection and enhance the capabilities of diagnosis, monitoring, management, emergency planning, and emergency response [56]. The main specific measures can be divided into the following points: Specific implementation options for the prevention and control of epidemics should be provided in light of the epidemic situation in specific countries and sectors.A complete monitoring system should be established. Disease monitoring and surveillance are key elements in the management of trans-boundary animal diseases (TADS), including ASF. This includes collecting, sorting, updating, and analyzing existing information related to epidemiology, which provide a knowledge base to understand the latest status of ASF in the region.The capabilities of diagnostic laboratories can be improved through the introduction of new diagnostic technologies and laboratory personnel training. Establish a network of diagnostic laboratories to predict the origin and evolution of disease outbreaks.Subregional management should be implemented. Domestic pigs in the ASF area must be placed in the enclosure and restricted to prevent contact with wild boars. In ASF-free zones, strict biosecurity measures will be carried out to protect domestic pig populations from diseases.

The main factors leading to the failure to control ASF in Africa include poor enforcement of rules and regulations, outdated legal frameworks, and lack of compensation mechanisms. Therefore, it is advocated to include an effective compensation mechanism in the national disease policy and legislation, as well as an increase in law enforcement efforts. The relevant departments of pig production need to obtain reliable information on the number and distribution of pigs, as this information is at least applicable to the identification and traceability at the herd level. Therefore, the strategy also recommends that pig identification and traceability should be implemented immediately in the commercial sector. Coordination, including information sharing and cooperation between countries and regions, is one of the most important and effective response tools to control ASF. Establishing consistency in the implementation of measures can avoid conflicts, repetitions, and wrong practices.

### 4.2. Europe

Since the case of Georgia in 2007, ASFV strains have flowed into eastern Europe, and ASF has spread rapidly to European countries, which has attracted the attention of European Union countries. They tried to stop the spread of the disease; however, few effective results have been achieved so far. ASF continues to spread to neighboring countries, mainly through contact with wild boars; other modes of disease transmission may occur at any time. With the continuous spread of ASF, EU member states have revised their emergency plans and implemented more protective measures, including disinfection of vehicles, stricter control of transit areas, strengthening of biosafety management on farms, and stricter vigilance programs [57]. ASF is a mandatory reported disease in the European Union, so any suspected ASF disease must be reported to the competent authorities. When the competent authority believes that it cannot be ruled out that ASF exists in a pig farm, it should immediately place the pig farm under official surveillance and conduct an inventory of all types of pigs in the pig farm. After the official diagnosis is ASF, the competent authority should immediately divide the protection zone and the monitoring zone. The protection zone should be established within a radius of at least 3 km around the outbreak site, and the monitoring zone should be within a radius of at least 10 km. After the pigs at the outbreak point have been culled, the buildings used for the placement of pigs, vehicles for transporting pigs or their carcasses, should be thoroughly cleaned and disinfected, as well as feces, floors and equipment that may be contaminated. Semen, eggs or embryos of pigs are prohibited from leaving the protection zone, and pigs are prohibited from moving and transporting on public or private roads. Experiments have shown that ticks from previously infected farms may contain infectious viruses for at least five years and three months after removing the infected host [58]. According to the current EU legislation, pig farms can be restocked within 40 days after the outbreak of ASF in the absence of pathogen vectors. If the vector is considered to be related to transmission, the minimum quarantine period is extended to 6 years. The inefficiency of trade meat inspection services, lack of supervision in the trade chain, and no logistics tracking increase the risk of ASF spread, which requires strengthening the supervision of the pig and pork trade.

Since ASF spread to Poland in 2014, it has caused a large number of wild boar and domestic pig infections. There are currently no confirmed cases of wild boar contact with domestic pigs in Poland, which means that wild boars have a limited role in the spread of ASF [59]. In Poland, most domestic pig infections occur in backyards and small farms. In order to achieve the eradication of ASF, the low biological safety of small farms is a problem that must be solved [60]. In contrast to Estonia, approximately six percent of the domestic pig population in Estonia is located in small farms, and most of the domestic pigs are concentrated in large farms [60]. Since February 2019, there have been no new cases of ASFV infection in Estonia. Estonia officially announced in 2020 that the country’s domestic pigs are ASF-free [61].

### 4.3. South America

Since Cuba first announced the ASF outbreak in 1971, it has spread to the Caribbean and South America, including the Dominican Republic, Brazil, and Haiti. However, the outbreaks in these countries were quickly controlled and eradicated. These countries have adopted strict control measures to eliminate the epidemics, such as isolation, culling, strict quarantine, regular cleaning and disinfection, and restricting the circulation of pig-related commodities [36]. The United States Animal Health Association, the United States National Pork Producers Council, and the National Association of State Department of Agriculture jointly drafted and launched an eradication plan in Haiti in April 1981. The plan is divided into four stages: publicity and education, slaughter/compensation, cleaning and disinfection, and establishment of pig sentry [62]. When ASF broke out in southern Brazil in 1978, the local government immediately took emergency control measures. During the emergency stage, the spread of the epidemic was mainly controlled by culling infected pigs. In November 1980, the Brazilian Government launched a large-scale national program to eliminate ASF and classical swine fever (CSF). It can be divided into three stages: the attack stage, consolidation stage, and maintenance stage. During the attack stage of the program, the main tasks were to complete the reorganization of the regional laboratory, the training of personnel, and the enhancement of ASFV diagnostic capabilities. The purpose of the consolidation stage and maintenance stage is to continue epidemiological surveillance in the country [63]. In addition, the mutual communication and joint cooperation between the government and farmers are indispensable factors for the eradication of ASF [64].

## 5. Diagnosis of ASF

### 5.1. Clinical Features and Pathological Changes

ASF is one of the most important viral diseases and seriously damages both domestic and wild boars. Clinical symptoms and pathomorphological changes vary widely depending on the virulence of different strains and the host immunity [65]. Several disease forms have been observed in domestic pigs and wild boars, including peracute, acute, subacute and chronic. 

Peracute ASF is usually induced by a highly virulent strain of ASFV, leading to sudden death in 1–4 days after the onset of clinical symptoms in domestic pigs. It is mainly characterized by high fever in animals (body temperature reaches 41–42 °C), loss of appetite and inactivity, dyspnea and skin congestion, as well as usually no obvious lesions in the organs [66].

Acute ASF induced by highly or moderately virulent virus strains is the most common form of disease, causing almost 90–100% of domestic pigs to die within 7 days. Its main symptoms are loss of appetite, shortness of breath, high fever (40–42 °C), constipation or diarrhea, tendency to cluster together, and miscarriage that may occur in pregnant sows. Large areas of subcutaneous hemorrhage can be observed in pigs, with erythema or bluish-purple appearance on the skin of ears, tail, distal end, chest, and abdomen [67]. Extensive necrosis and hemorrhage of lymphatic tissue and pulmonary edema are typical characteristics seen in the infected pigs. Bleeding occurs in the medulla of the lymph nodes, which is the reason for the appearance of marbling. In addition, the spleen is friable, dark-red to black, and enlarged, which may be five times larger than normal, accompanied by peripheral necrosis. Bleeding spots appear on the surface of most internal organs, while petechiae appear on the surface of the heart (epicardium), urinary bladder, and kidneys (on the cortical and renal pelvis) [65,68,69].

This form of subacute ASF is mainly caused by moderately virulent isolates, causing deaths within 7–20 days of infected pigs, with lethality rates ranging from 30% to 70%. The clinical features of the subacute form are similar to acute ASF, with generally less intense clinical symptoms. However, the bleeding and edema observed in the subacute form of ASF is more severe than those reported in the acute forms, which may be the cause of leukopenia and thrombocytopenia caused by ASFV infection [70,71,72]. Systemic organ hemorrhage and edema are the main clinical manifestations of subacute ASF. In some cases, we can observe that the joints of infected pigs are often swollen due to the accumulation of fluid and fibrin. Pathological changes of organs include extensive subcutaneous hemorrhage, necrosis of lymph nodes, pulmonary edema, splenomegaly, and hemorrhage of kidney, mesenteric and serous membrane. Miscarriages are more frequent in pregnant sows [44].

Low virulence isolates lead to chronic form of the diseases with low mortality and no vascular damage. However, signs of growth retardation, weight loss, low mortality, joint swelling, and skin ulcers may be observed. This kind of infected domestic pig carries the virus in its blood for a long time, which is accompanied by the risk of spreading the virus [44,73].

If ASF is diagnosed only by clinical features, it is easy to be confused with other porcine hemorrhagic diseases, including classical swine fever (CSF), high pathogenic porcine reproductive and respiratory syndrome (HP-PRRS), and swine erysipelas. Regardless of clinical examination or autopsy, it is difficult to distinguish ASF. These diseases should be considered in the differential diagnosis of acute febrile hemorrhagic syndrome of pig. Therefore, laboratory tests are essential for the identification of ASF.

### 5.2. Etiological Diagnosis

#### 5.2.1. Virus Isolation and the Hemadsorption (HAD) Test

The effectiveness of all diagnostic methods developed so far, including polymerase chain reaction (PCR), enzyme-linked immunosorbent assay (ELISA), and Western blotting (WB), depend on the concentration and quality of virus samples collected from infected areas. If the samples are poorly obtained or stored, the accuracy of the diagnosis may be affected. Therefore, appropriate sensitive culture cells are required to isolate the virus. It is essential to obtain virus stocks for further molecular and biological research. Some studies have shown that ASFV replicates in the cells of the mononuclear phagocyte system of infected pigs, mainly monocytes and macrophages [74,75]. After entering the animal body, ASFV mainly replicates in the mononuclear phagocytes of the tonsils or lymph nodes, and spreads to the secondary replication organs, such as liver and kidney, through the lymphatic system and blood circulatory system [76]. We can isolate ASFV strains from the blood (EDTA), spleen, liver, lymph nodes and tonsils of infected pigs for further laboratory diagnostic tests [77]. On the other hand, if the preservation quality of the virus samples is poor, the virus needs to be replicated in susceptible cells derived from the blood or lungs (alveoli) of healthy pigs. After the virus is isolated and cultured, the HAD test is performed.

Primary peripheral blood mononuclear cells and alveolar macrophages are the main targets for ASFV infection and passage. During the cultivation process, they are closer to the infection state in the organism, which is more credible for the study of natural infection processes. Macrophages do not divide in vitro, and during the process of ASFV infection on porcine macrophages, viral DNA is mainly synthesized [78]. However, the shortcomings of the inability to form a subculture and complicated cell extraction process limit large-scale virus isolation and diagnostic tests. In the beginning, these problems were initially overcome by establishing an African green monkey kidney cell line, such as VERO cells or Monkey Stable cells (MScs), which were adapted to some ASFV strains [79,80]. There are reports describing COS-1 cells as an established cell line sensitive to all ASFV isolates tested, which can be used for diagnosis, detection and virus amplification [81]. In addition, the mature cell lines derived from porcine alveolar macrophages, such as IPAM and WSL, are closer to the natural environment and more accurately simulate the ASFV infection process in vivo [77]. The WSL cells are more suitable for the study of ASFV protein expression and virus production compared with the IPAM cells, whereas the level of infection and virus production in WSL cells is lower than in porcine alveolar macrophages (PAMs) [82].

The HAD test can specifically detect ASFV based on its special function for red blood cells. Red blood cells are allowed to attach to infected macrophages and peripheral blood mononuclear cells, leading to rosette changes before the cytopathic effect [83]. The HAD test is more sensitive to ASFV detection than serum antigen detection and is often used for routine sample detection. However, it takes a few days to obtain results and relies on regular acquisition of primary cells prepared from fresh pig tissue. In addition, the emergence of nonhemadsorbing ASFV strains increases the possibility of false negative results in this test [84,85,86].

#### 5.2.2. Polymerase Chain Reaction (PCR) Assay

The conventional PCR assay is one of the conventional nucleic acid detection methods recognized by OIE and widely used in laboratory diagnosis of ASFV. This assay has the advantages of simplicity, speed, sensitivity, specificity, and low requirements for specimen purity. It is usually applied to the detection of ASFV genomes in serum, blood or organ samples. Based on the comparison of the nucleotide sequences of gene VP73 (a part of gene VP72) of seven different representative ASFV strains, Aguëro et al. designed specific primers and established a novel ASFV PCR assay, which was recommended by the OIE Diagnostic Test and Vaccine Manual for Terrestrial Animals, 2012 [87]. It showed higher sensitivity than that recommended by OIE Diagnostic Testing and Vaccine Standard Manual, 2010 (OIE, 2010) and is suitable for detecting almost all ASFV strains worldwide [88]. A novel PCR assay based on the highly conserved region of the VP72 sequences of all ASFV strains showed higher sensitivity than the OIE-validated PCR assays [89]. Ticks play an important role in the spread of ASF, and the development of detection methods for ASFV in ticks has a positive effect on disease prevention and control. A nested PCR assay, with an internal control, was established to detect ASFV DNA in ticks (*Ornithodoros* species) in the wild [90].

The principle of multiplex PCR is almost the same as that of conventional PCR. It is able to detect multiple viruses without cross-reactivity at the same time, which has important guiding significance for detecting multiple virus infections in pigs [91]. Erickson et al. established a novel accompanying automated electronic microarray assay for simultaneous detection of seven diseases in pigs. Its rapid and high-throughput detection characteristics indicate its potential huge application prospect [92]. Based on the Bio-Plex suspension array system, a multiplex PCR was established to detect seven pathogens, including ASFV. First, the multiplex PCR amplification is performed, and then the amplified products are coupled with specific probes on the flow beads. Finally, the Bio-Plex system is used for detection. The Bio-Plex system identifies each specific reaction based on the color and fluorescent signal of the beads, and the change of the median fluorescence intensity (MFI) is regarded as the basis for diagnostic analysis [93]. Each magnetic bead is used as a separate detection body, which can carry out a large number of biological tests at the same time, and truly realizes high throughput and high speed. This is of great significance to the development of liquid chip technology for disease diagnosis, which is a development direction of future diagnostic technology. However, the low detection rate and expensive instrument costs are urgent problems that need to be solved. The main features of the above detection methods are summarized, and a comparison of conventional PCR detection methods is shown in Table 2.

PCR provides a sensitive, specific and rapid method for the detection of ASFV, which is more sensitive and specific than antigen detection methods such as ELISA. However, the extreme sensitivity of PCR assays can also make them prone to cross-contamination, leading to false positive results. Due to the presence of inhibitors or damage nucleic of acids during nucleic acid extraction, false negative PCR results cannot be ruled out [94].

#### 5.2.3. Real-Time PCR Assay

Real-time PCR is the first choice for pathogen detection and the gold standard for ASFV detection. In fact, real-time PCR is considered to be the most sensitive and reliable method realized so far. Real-time PCR is described as a technology that specifically recognizes target sequences through the fluorescent signal of oligonucleotide probes and detects gene amplification in real time [87].

The first report for a real-time PCR assay for the detection of ASFV was developed by King et al., which is described in the OIE, 2012. In this assay, amplification primers for gene VP72 of ASFV are designed, and the PCR amplicons are detected by the 5′-nuclease assay system. An artificial mimic is designed based on the two-color TaqMan probe to identify post-PCR products. The successful amplification of the mimic indicates that there is no substance that inhibits the polymerase chain reaction, thus verifying the negative result [95]. In order to improve the accuracy of detection, applying an internal control is an important aspect of quality control. To prevent false negative results in the presence of PCR inhibitors, an internal control PCR assay within the gene endogenous β-actin was developed for the first time. The experimental results showed that the new method improves sensitivity and reduces the rate of false negatives, compared to the method of King et al. [96]. Fernández et al. established an internal control PCR method that combines commercial universal probe library (UPL) probes within the endogenous gene β-actin with specially designed primers to amplify the target gene VP72. While ensuring high sensitivity, UPL probes take less time and save about 30% in costs compared to customized standard hydrolysis probes [97]. Wang et al. established a quantitative real-time PCR assay combined with lyophilized powder reagent (LPR), including specific primers and specific TaqMan molecular probes. This assay reduced the detection cost and effectively shortened the diagnosis time of ASFV and was suitable for laboratory diagnosis [98].

The principle of DNA biosensors is to combine single-stranded DNA molecules with known nucleotide sequences fixed on the surface of the sensor or transducer probe. The conjugate obtained hybridizes with another complementary ss-DNA molecule to form a double-stranded DNA, showing a certain physical signal [99]. In recent years, reports have proposed a biosensor-based method for detecting ASFV nucleic acid in pig blood. This method uses locked nucleic acid (LNA) with a single-stranded DNA probe as a complementary recognition element for the gene VP72. This method allows the rapid quantification of ASFV nucleic acid to achieve the purpose of detection. In fact, the combination of biosensors in PCR detection has major advantages of real-time and label-free detection. However, it cannot directly detect blood samples, unless the DNA is extracted, for the reason that certain substances in the blood may react specifically with the sensor surface [100]. Liu et al. developed a duplex real-time PCR assay on a portable instrument for rapid detection and identification between ASFV and CSFV. The measurement is performed on a portable, battery-powered PCR thermocycler T-COR4, which provides convenience for testing. However, it has the problem of low sample throughput, low sensitivity, and false negatives. In addition, the T-COR4 assay may not be able to accurately identify infected pigs with low viremia [101]. The accuracy and test conditions of the portable thermal cycle still need to be further improved and optimized in order to further improve the practical applicability of the assay. As an on-site diagnostic tool, it provides the possibility and practicality of early on-site screening. Table 3 summarizes and shows the comparison of the above real-time PCR detection methods.

#### 5.2.4. Isothermal Amplification Technologies: RPA, LAMP and CPA

##### Recombinase Polymerase Amplification (RPA)

PCR is the most widely used method of DNA amplification for detecting and identifying infectious and genetic diseases and for other research purposes. It possesses lots of advantages; however, it requires a thermocycling machine to separate double-stranded DNA and amplify the desired fragments. Isothermal amplification methods were developed based on some new studies in molecular biology about DNA synthesis and some accessory proteins that assist nucleic acid amplification. This technology eliminates the dependence of amplified DNA on a thermocycling machine equipment and expands the use environment of diagnostic tests [102]. Several common isothermal technologies for diagnosing ASFV are as follows, such as recombinase polymerase amplification (RPA), loop-mediated isothermal amplification (LAMP), and cross-priming amplification (CPA).

RPA is an isothermal amplification experiment based on recombinase developed by the British company TwistDx, which is sensitive, rapid (5–20 min), and simple to operate [103]. After the RPA starts to react, the recombinase is first combined with the primer to form a complex. The complex searches for the homologous double-stranded DNA and undergoes a strand exchange reaction with it, forming a D-ring structure, which binds the single-strand DNA binding protein (SSB) to the DNA strand. Then, the DNA synthesis starts, and the DNA is amplified to form two new double-stranded DNAs, so that the target region on the template is amplified exponentially [104]. The amplified products can be read by agarose gel electrophoresis, real-time fluorescence, or lateral flow chromatography test strips.

Wang et al. established a rapid and specific detection method for ASFV based on RPA. The RPA reactions were performed using a TwistAmp exo kit (TwistDX, Cambridge, UK), and were run on the Genie III scanning device, which was applied for the detection of ASFV within 20 min [105]. RPA is universally specific to all ASFV genotypes and has no cross-reactivity to other pathogens [106]. Some studies have proved that RPA can be used in conjunction with lateral flow detection (LFD) to observe the results in a more intuitive way. The clinical results show that they have close sensitivity, compared to OIE real-time PCR. It is worth noting that the test tube must be opened only after the amplification finishes because it may carry contaminants from the fields [107]. The CRISPR/Cas system has the ability to accurately identify specific sequence. Based on the combination of CRISPR/Cas12a with recombinase assisted amplification (RAA) and immunochromatographic lateral flow strips, a test strip method for ASFV field detection has been developed [108].

##### Loop-Mediated Isothermal Amplification (LAMP)

LAMP is a novel nucleic acid amplification technology suitable for genetic diagnosis, published by Japanese scholar Notomi in 2000. Four specific primers were designed for the six regions of the target gene. Under the action of strand displacement DNA polymerase, the nucleic acid could be amplified at a constant temperature of 60–65 °C, at about 15–60 min [109]. LAMP possesses the characteristics of simple operation and high sensitivity. James et al. established a LAMP assay for detecting ASFV, targeting the ASFV topoisomerase II gene. This assay also demonstrates that the LAMP amplicons can be detected reliably with good sensitivity using LFD [110]. In order to better cope with the conditions on the scene, researchers continue to improve the detection methods, trying to express the results of experiments through visible changes. For example, Tran et al. established the real-time LAMP assay and the visual LAMP assay using pH-sensitive dyes to develop color, which directly detected ASFV from serum samples [111]. Yu et al. combined image processing and a hue saturation value (HSV) color model with LAMP assay to display the detection results in the form of images [112]. Zhu et al. combined the hive-chip with the direct LAMP assay to establish a diversified and visual detection platform [113]. The number of tested samples and the types of strains involved in these reports are limited, and the experimental results need to be further verified. In addition, when aerosol pollution occurs, it is prone to a high false positive rate [114]. They can only be used as an early diagnosis of suspected infected areas, not as the final judgment result. In short, LAMP technology has development potential, but it still needs to be further optimized.

##### Cross-Priming Amplification (CPA)

The CPA assay is a novel nucleic acid isothermal amplification technology, which does not require an initial denaturation step, independently developed by Ustar in 2008. There were 4–5 primers (including one or more cross-primers) designed to amplify the target gene through strand displacement at 63 °C [115]. The CAP assay has high sensitivity, specificity, and a wide range of applications. It can not only detect viruses in conventional blood and serum, but also detect and classify meat varieties [116,117]. Fraczyk et al. reported a cross-primer amplification (CPA) assay, which was the first development and application of direct blood and serum testing of pig and wild boar. It tested 86 animal serum samples and CPA had equal sensitivity, in comparison to the official UPL real-time PCR [118]. The main characteristics of these assays are summarized in Table 4.

### 5.3. Serological Diagnosis

Serological diagnosis is the most commonly used diagnostic detection method because of its simplicity, relatively low cost, and low requirements for specialized equipment. There is currently no vaccine against ASFV, which means that the presence of ASFV antibodies always indicates infection. Therefore, antibody testing is particularly important for the diagnosis of ASF. Currently, serological testing for ASFV includes antigen testing and antibody testing, but the conditions for antigen testing are more stringent. Due to the fact that ASFV cannot completely neutralize antibodies, antibody detection has gradually become the mainstream serological diagnosis method. The ASFV antibody-based test recommended by OIE is ELISA for antibody screening, supplemented by other confirmatory tests, such as immunoblotting tests (IBTs), indirect fluorescent antibody (IFA) tests, and indirect immunoperoxidase tests (IPTs) [87].

#### 5.3.1. Antigen-Based Detection

##### Fluorescent Antibody Test (FAT)

FAT is one of the commonly used antigen detection methods and is able to identify nonhemadsorbing ASFV strains. The test has the advantages of rapidness, high sensitivity and specificity, as well as high detection rates for acute infections. The organ material from a pig suspected of ASFV infection was pressed onto a smear or thin frozen section. After the intracellular specific antibody is combined with fluorescein isothiocyanate (FITC), the virus antigen can be detected under the microscope in the form of fluorescent inclusion body or particle [120]. FAT is less sensitive to subacute and chronic ASF. It is possible that the antigen–antibody complex in the process of chronic infection competitively blocks the combination of ASFV antigen and antibody, resulting in a false negative [120]. Therefore, it is generally only used as an auxiliary detection method. In addition, FAT is usually used for ASFV in vitro cell culture, virus localization, replication detection and other biological research.

##### Direct ELISA

Direct ELISA based on monoclonal antibody coatings can directly detect ASFV antigens, but it is only recommended for acute cases. For subacute and chronically infected animals, the sensitivity of antigen ELISA is significantly reduced. This may be related to the production of the antigen–antibody complex in infected pig tissues [121]. Wardley et al. established a direct ELISA by antisera to detect ASFV virus antigens and detected a limit antigen concentration of 50–500 HAD_50_/mL [122]. The successful screening of monoclonal antibodies provides a basis for the development of direct ELISA. Later, a sandwich ELISA based on VP73 monoclonal antibody was established and proved to be highly sensitive to the detection of homologous antigens, but there are no on-site diagnostic data [123]. A comparison of two direct sandwich ELISAs was described by Hutchings et al.—One coated with polyclonal serum coating and the other coated with a combination of monoclonal and polyclonal sera. This study proved that the detection method using polyclonal antiserum is slightly more sensitive than the detection method using monoclonal antibodies [124]. Sastre et al. established a lateral flow assay (LFA) for antigen detection. This test is based on a monoclonal antibody against protein P72 of ASFV as a test line capture agent, and its sensitivity is similar to commercial antigen-ELISA. This assay can detect ASFV infection early and is especially suitable for large-scale field screening and testing [125].

#### 5.3.2. Antibody-Based Detection

ASFV infection will cause a strong humoral immune response in pigs and last for a long time. However, the antibodies produced cannot be completely neutralized, which provides the possibility of antibody detection [126]. When pigs are infected with attenuated or low virulence isolates, serological testing may be the only way to detect infected animals.

##### Indirect ELISA

The immunoelectroosmophoresis (IEOP) test was one of the earliest serological assays used in the laboratory to diagnose ASFV. The IEOP test was superior to the complement-fixation tests and agar gel diffusion test in detecting ASFV antibodies, but it required antigens prepared from ASFV-infected VERO cells extract [127]. The IEOP test is time-consuming, laborious, and not easily adapted to large-scale investigations. With the advantage of higher sensitivity and simpler operation, ELISA has replaced IEOP as the most suitable and commonly method for ASFV detection of large amounts of serum [122]. However, there is a problem of false negative diagnosis caused by the immediate death of animals or the existence of viral immune complexes after infection [128]. The soluble antigen protein of ASFV-infected cells is separated by sucrose precipitation centrifugation, and the supernatant above the sucrose layer is used as an ELISA antigen. This crude antigen is currently recommended as a test reagent in screening and diagnostic tests by OIE [87].

If more accurate diagnostic results are to be obtained by ELISA assay, more reliable solid phase carriers and purified antigens must be carried out [129]. Tabares et al. reported the preparation of a semipurified ASFV major capsid protein VP73, which greatly improved the reliability of ELISA as the coating antigen [130]. ASFV antibodies in serum samples were detected at least 2 days earlier by using crude antigen as opposed to semipurified P73. This shows that ASFV proteins P25, P25.5, and P30 seem to have better immunogenicity and earlier production of specific antibodies than semipurified P73 [131]. By screening the library with polyclonal antiserum in domestic pigs, two structural proteins P30 (also named P32) and P54 were clearly identified as high antigenicity during infection [132,133]. Oviedo et al. verified their detection effect in ELISA and immunoblotting test (IBT) by baculovirus expression of recombinant proteins P30 and P54 and suggested that P30 is more suitable as an antigen for ELISA. In contrast, P54 should be used as an antigen for the IBT. The immune properties of these proteins may be related to the difference between the number of linear and conformational epitopes of the proteins. In addition, the combined application of proteins P54 and P30 in serological diagnosis of ASFV can improve the sensitivity [133]. Filgueira et al. established an ELISA test based on the recombinant protein P30 by a baculovirus vector in insect larvae. It was more sensitive than the ELISA test recommended by OIE to detect ASFV specific antibodies in experimentally infected animals at the early stage of infection [134]. Gallardo et al. found that the ASF virus polyprotein pp62 has good antigenic properties, and the ELISA test based on the protein pp62 showed good reactogenicity. Especially when testing poorly preserved samples, the detection accuracy of pp62 is significantly better than proteins P30 and P54, and it is even not necessary to confirm the results by IBT [135]. Gallardo et al. evaluated the role of four recombinant proteins of ASFV (pA104R, pB602L, P54 and pK205R) as diagnostic ELISA tools for ASFV serological diagnosis of samples from Europe and Africa. The results of recombinant protein ELISA tests were consistent with those approved by OIE. Among them, serum samples from east Africa were highly specific, but unexpectedly low in terms of sensitivity [136]. The reason for the low incidence of detectable serum response to ASFV infection in east Africa was not clear [137]. An explanation was described that different porcine breeds may have different immunological pedigree, clone deletion, porcine leukocyte antigen type or innate immune mechanisms [134,138].

##### Indirect Fluorescent Antibody (IFA) Test, Indirect Immunoperoxidase Test (IPT), and Dot Immunobinding Assay (DIA)

The principle of IFAs is to infect a single layer of VERO cells with a cell-adaptable ASFV strain, and then couple the fluorescein to a specific antigen–antibody reaction to achieve the detection of ASFV antibodies. In the case of positive samples, specific fluorescence was observed in the cytoplasm of infected cells under a microscope. IFA is a rapid, highly sensitive and specific ASFV antibody detection technique that is suitable for the detection of antibodies in serum, plasma or tissue secretions [139]. The IFA test is more sensitive than the ELISA test and can detect ASFV antibody at the earlier serological reaction stage [137]. Especially in the early stage of infection, the IFA and IPT provide good choices for serological diagnosis when infected animals have almost no antibodies. The principle of IPT is similar to IFA, except that IPT uses the enzyme chromogenic system, which is easier to judge.

The detection serum contains antibodies against the cells themselves, and uninfected cells will also emit fluorescence. Therefore, confusion may occur when interpreting the results and wrong diagnosis may be reached. Pan et al. developed a method of indirect immunoperoxidase plaque-staining (IIPS). It has all the principle features of IPT and the results can be directly read with the naked eye under ordinary light, allowing a technician to perform 400 tests a day [129]. 

Pastor et al. developed a nitrocellulose test strip for the detection of ASFV on the basis of dot immunobinding assay (DIA), which was used to detect antibodies in the field. The cytoplasmic soluble antigen (CS-P) of ASFV, coated on the nitrocellulose test strip, reacts with serum samples. It can detect antibodies by the protein A–peroxidase conjugate. The working principle of DIA is similar to ELISA. In addition, due to the stability of nitrocellulose strips, there is no significant difference in chromaticity between freshly prepared nitrocellulose strips and nitrocellulose strips stored for 6 months [140]. Compared with other serological tests, DIA-CS-P has the advantages of simplicity, rapidity and low cost, and it is possible to screen pigs suspected of latent ASFV infection under field conditions.

##### Immunoblotting Test (IBT)

In recent outbreaks of ASF in Europe and Western Hemisphere countries, clinical mild and low mortality cases caused by low virulent strains are common [141]. Sometimes nucleic acid testing cannot detect the antigen, and so serological detection of antibodies is a good choice. IBT is a method that can replace IFA to detect ASFV antibody, and its sensitivity is even better than the IFA. Pastor et al. used pig antiserum for immunoblotting experiments and observed band reactions of IP12, IP23.5, IP25.5, IP30 and IP34 [139]. In order to avoid the false positive reaction caused by the cellular protein contained in the antigen used in traditional IBT, a novel IBT for detecting ASF virus antibody was established by using the recombinant virus protein P54 expressed in *E. coli*. The production costs of reagents for recombinant Western blot are cheaper than traditional Western blot, and there is no contact with an infectious virus in antigen production. Secondly, there are fewer bands to display the results, thus allowing accurate and clear viewing of the experimental results [142]. The unpurified recombinant protein and the protein expressed by prokaryotic cells can easily affect the experimental results. Kazakova et al. linked protein P30 with 6xHis tag and purified to high purity recombinant protein P30 by chromatography and established an immunoblotting test system for serological diagnosis of ASFV. The specificity and sensitivity were improved [143].

Each diagnostic technique has its own distinct characteristics, providing a variety of options for ASFV diagnosis. Some technologies have high sensitivity and specificity but may be limited by expensive experimental equipment such as PCR and FAT. Some diagnostic techniques, such as ELISA, have inferior sensitivity and specificity but lower costs. Table 5 summarizes the characteristics of several conventional experimental techniques for ASFV.

## 6. Discussion and Conclusions

Since ASF was first described in the early 20th century, ASFV has been circulating in different parts of the world, causing serious losses to the pig industry and the world economy. As there is currently no available vaccine, the prevention and control of ASF have become the most urgent and difficult problem to solve. Currently, the primary means of prevention and treatment of ASF depend on strict biosecurity control. Early detection, early reporting, and early disposal are important means to control and eradicate ASF. In recent years, ASFV diagnostic methods have been developed in various ways and have shown good diagnostic effects. Accurate and rapid diagnosis of ASFV makes it possible to take early control measures to curb the development of epidemics.

Diagnosis based on clinical symptoms can only be used as initial diagnosis for ASFV infection and is far from being the final diagnosis. The HAD test is a highly specific and sensitive detection method. However, the emergence of nonhemadsorbing ASFV strains and the long detection time prevent the HAD test from becoming a routine detection method. PCR and real-time PCR are still the most reliable methods for diagnosing ASFV infection, among which real-time PCR with high sensitivity is regarded as the gold standard. The development of isothermal amplification technology eliminates the constraints of expensive laboratory equipment, but the sensitivity and effectiveness still need to be verified by a large amount of clinical diagnostic data. The T-COR4 test developed by the combination of a portable, battery-powered PCR thermal cycler and real-time PCR may be the future development direction of herd-based detection for ASFV. The combination of isothermal amplification technology and CRISPR-Cas12a also has good development potential. ELISA has the advantages of rapid detection, simple operation, and relatively low cost, and is the most widely used method for ASFV detection. ELISA is suitable for acute cases, and the detection rate for chronic cases is not high. Therefore, for samples that are shown to be positive by ELISA, it is recommended to confirm by nucleic acid detection, immunoblotting or indirect fluorescent antibody test. 

The current commercial ELISA kits are mainly based on proteins P30, P54, P72, and pp62, among which protein P30 is considered to be the most suitable for immunoadsorption tests. At present, lateral flow devices (LFDs) based on ASFV antigen detection have been developed. The use of this new type of pen-side test provides first-line diagnosis in emergency situations. The sensitivity of this test is limited, and it needs to be used in conjunction with laboratory diagnosis. This rapid and simple detection method can be used for early on-site detection of ASFV infection, for places where laboratory equipment is very simple or lacking. Lateral flow detection does provide convenience for diagnosis, and how to improve sensitivity is the key to the development of this technology. In addition, isothermal amplification technology is a nucleic acid amplification technology emerging in the 21st century. This technology basically meets the current demand for ASFV detection in the field and has great application prospects for on-site diagnosis. However, this method is a single-channel detection, and it is difficult to achieve simultaneous detection of hundreds or thousands of samples. In view of this, multichannel rapid detection—for example, by combining with electronic chip technology—will become the future development direction of this technology. 

## Figures and Tables

**Figure 1 vaccines-09-00343-f001:**
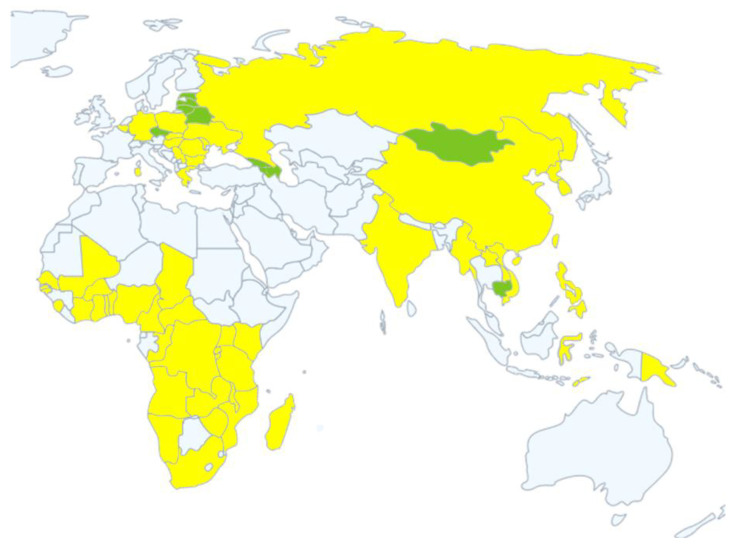
The distribution and recent status of African swine fever (ASF) in Africa (2016–2020), Europe (2007–2020), and Asia (2018–2020). Yellow represents continuing outbreaks, green represents resolved outbreaks. Source: OIE WAHIS African Swine Fever (ASF) [46].

**Table 1 vaccines-09-00343-t001:** Current status of ASF in Europe and Asia from 2007 to December 2020.

Country	Year or Date Reported	Status
Georgia	2007–2008	Resolved
Armenia	2007–2008, 2010–2011	Resolved
Russian Federation	2007	Continuing
Azerbaijan	2008	Resolved
Latvia	2014–2019	Resolved
Estonia	2014–2019	Resolved
Ukraine	2012, 2014	Continuing
Belarus	2013	Resolved
Lithuania	2014–2019	Resolved
Poland	2014	Continuing
Moldova	2016	Continuing
Hungary	2018	Continuing
Czech Republic	2017–2018	Resolved
Romania	2017	Continuing
Belgium	2018	Continuing
Bulgaria	2018	Continuing
Slovakia	July, 2019	Continuing
Serbia	2019	Continuing
Greece	February, 2020	Continuing
Germany	September, 2020	Continuing
Sardinia	2007–2020	Continuing
China	3 August 2018	Continuing
Democratic People’s Republic of Korea	23 May 2019	Continuing
Lao People’s Democratic Republic	20 June 2019	Continuing
Myanmar	1 August 2019	Continuing
The Philippines	25 June 2019	Continuing
South Korea	17 September 2019	Continuing
Vietnam	19 February 2019	Continuing
Mongolia	15 January 2019	Resolved
Cambodia	2 April 2019	Resolved
Timor-Leste	9 September 2019	Continuing
Indonesia	September, 2019	Continuing
Papua New Guinea	March, 2020	Continuing
India	May, 2020	Continuing

Source: OIE WAHIS African Swine Fever (ASF) (2007–2020) [46].

**Table 2 vaccines-09-00343-t002:** Conventional Polymerase Chain Reaction (PCR) diagnostic comparison for the detection of African swine fever virus.

Detection Method	Target Gene	Internal Control	Display of Results	Sample Type	Limits of the Test	Validation		Reference
						No. of Diverse ASFV Isolates	No. of Clinical Samples	
PCR	VP73	None	Agarose gel electrophoresis	Tissue, blood, serum	0.12 HADU_50_	22	70, including 18 serum, 4 blood	[88], OIE
PCR	VP72	None	Agarose gel electrophoresis	Blood	60 copes	14 (4 genotypes)	62 blood	[89]
Nested PCR	VP72	Foreign DNA	Agarose gel electrophoresis	Ticks supernatant	0.49 CPE_50_	20 (7 genotypes)	60 ticks	[90]
Multiplex PCR	VP72	None	Agarose gel electrophoresis	Tissue	1.50 × 10^3^ copes	Gene fragments	10 pigs, 76 wild boars	[91]
Multiplex PCR	VP72	None	Fluorescent intensity (FI) data	blood	10 copes/μL	4 (2 genotypes)	24	[92]
Bio-Plex multiplex PCR	VP72	None	Bio-Plex	Tissue, blood	10^3^ copes	Gene fragments	137	[93]

**Table 3 vaccines-09-00343-t003:** Real-time PCR diagnostic comparison for the detection of African swine fever virus.

Detection Method	Target Gene	Internal Control	Probes	Sample Type	Ct Value	Limits of the Test	Validation		Reference
							No. of Diverse ASFV Isolates	No. of Clinical Samples	
Real-time PCR	VP72	Artificial template (mimic)	TaqMan	Tissue, ticks, cell lines	21.79–30.59	10–100	25 (9 genotypes)	None	[95], OIE
Real-time PCR	VP72	β-actin	TaqMan	Tissue, blood, serum, ticks,	15–30	5.7–57	44 (7 genotypes)	281	[96]
Real-time PCR	VP72	β-actin	UPL	Tissue, blood, serum, ticks, cell lines	18.79–35.56	below 18	46 (19 genotypes)	260	[97], OIE
Real-time PCR (LPR)	VP72	None	TaqMan	Blood	<32	100 (plasmid)	Gene fragments	218 (in China)	[98]
Real-time PCR (LNA)	VP72	None	ssDNA/LNA	Tissue, blood	None	178	14 (2 genotypes)	20	[100]
Real-time (T-COR4)	VP72	None	None	Tissue, blood, serum	34.3	10,000	Gene fragments	41	[101]

**Table 4 vaccines-09-00343-t004:** Recombinase polymerase amplification (RPA), loop-mediated isothermal amplification (LAMP) and cross-primer amplification (CPA) diagnostic comparisons for the detection of African swine fever virus.

Detection Method	Target Gene	Reaction Time	Reaction Temperature	Display of Results	Probe	Limits of the Test	Validation		Reference
							Compared withrPCR (OIE)	No. of Samples	
RPA	VP72	20 min	39 °C	Genie III scanner device	Exo	100 copes	Consistent	40	[105]
RPA	VP72	10 min	38 °C	LFD	FITC	150 copes	Consistent	145	[107]
RAA-CAS12a	VP72	1 h	37 °C	LFD	CAS12a	1 × 10^−9^ M	Decreased sensitivity	None	[108]
LAMP	Topoisomerase II gene, VP73	25 min	64–66 °C	LFD	None	330 copes	Consistent	42 (from 7 pigs)	[110]
LAMP	Topoisomerase II gene	30 min	60 °C	Direct colorimetric	None	10 HAD_50_/mL	Consistent	97 (from Vietnam)	[111]
CPA	VP72	45 min	56.2 °C	SYBR Green I	LightCycler 480	7.2 copes (plasmid)	Consistent	10 pigs, 76 wild boars	[118]
CPA	VP72	60 min	59 °C	LFD	Biotin	200 copes	Consistent	65	[119]

**Table 5 vaccines-09-00343-t005:** Comparison of laboratory diagnostic techniques for African swine fever.

Detection Method	Time	Sensitivity	Specificity	Sample Type	Cost	Comments
PCR	5–6 h	XXX	XX	Tissues, blood, ticks or cell cultures	$$	Most common method Susceptible to contamination Detects live or dead virus
HAD	7–21 days	XX	XXX	Porcine macrophage cells	$$$$	Gold standard, only used in a few reference laboratories
FAT	75 min	XXX (for early detection)	XXX	Cryostat sections, impression smears, cultured cells	$$$	Recommended when PCR is unavailable or lack of experience Needs a fluorescent microscope Lack of sensitivity after the first week post-infection
ELISA	3 h	X (for early detection)	X	Serum, macerates	$	Screening test In-house and commercial kits available
IBT	3 h	X	X	Serum	$$$$	Confirmatory test No commercial kits
IFA	4 h	XXX	XX	Tissue exudates, serum or plasma	$$$	Confirmatory test No commercially available reagents Needs a fluorescent microscope

X, XX, and XXX represent the levels of sensitivity and specificity. $, $$, $$$, and $$$$ indicate the level of cost. Source: FAO [67].

## Data Availability

Data sharing not applicable. No new data were created or analyzed in this study. Data sharing is not applicable to this article.

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
