# Peer review of "Development of Diagnostic Tests Provides Technical Support for the Control of African Swine Fever"

_vaccines, 2021, doi:10.3390/vaccines9040343_

Round 1

Reviewer 1 Report

the paper written by Qiu and collaborators is well written and treatted a very important issue. However the bibliography needs to be updated:

paragraph 1:line 58-61 there is an error. Delected causing to...
move .... At the present until ... on the market.
Insert some references on escape mechanisms e.g IFN

paragraph 2:lines 81-89 there are recent studies conduced in Sardinia on 2020
and on the control eradication strategy in Europe. I have to be entered
e.g. Loi et al., 2020 Vaccine.
4.2 lines 184-213. Insert studies condused in estonia and poland  

Author Response

Thank you for your valuable comments and suggestions. Based on your comments and suggestions, we make the following changes:

response:

paragraph 1: causing to...  has been delected, (lines 66-67) 

At the present until ... on the market. has been modified to :Currently, there is no vaccine against ASF. (lines 61)

some references about IFN and inhibiting apoptosis have been inserted. (lines 61-64)

paragraph 2: recent studies conduced in Sardinia in 2020 and on the control eradication strategy in Europe have been added (lines 108-115)

4.2: studies condused in estonia and poland have been added (lines 264-273)

Reviewer 2 Report

The work presented is an interesting review of the current situation about african swine fewer virus (ASFV) mostly at the diagnosis level. The work is well written and organized and represent a useful starting point for anyone who want an introduction about ASFV. I detected some low numbers of small typos, so I suggest a last revision on the entire paper. Finally, I strongly advise to add a 4.3 small paragragh about the ASFV situation in Americas, since some countries were interested by this virus in the past.

Author Response

Thank you for your valuable comments and suggestions. Based on your comments and suggestions, we make the following changes:

We have corrected typos in the entire manuscript.

We have added a short paragraph describing the eradication of African swine fever in the Americas in 4.3 (lines 274-295)

Reviewer 3 Report

The authors review the status, transmission, control and diagnosis of ASFV.  However, the manuscript suffers from poor English, frequent typos and lack of thorough proofreading, at times making it difficult to read and/or understand.  The document requires major editing and proofreading; too many things to list here.

Some items to note:

Line 59 and 79: Gene types should be genotypes

Oriniformes and orniformes mollusc???

Line 124-5: this sentence is not clear; needs to clarify ornithodoros soft ticks are vectors, other insects can be mechanical transmitters.

Line 49: Africa misspelled

Many sentences in section 4 are incomplete and read as bullet points. This happens several other places throughout the manuscript as well.

ASF or ASFV is sometimes defined, sometimes not, (also happens with CSF and PCR etc.) and sometimes referred as AFV, AFS or African classical swine fever used throughout the manuscript eg. Line 165, 208, 609, 612

Line 274 – 271 sentences are repeated

Line 296: HAD test more sensitive than viral DNA?

Line 299: non-hemosucking?

Line 315: detect ticks?

Define items in the table 1;  why not table summarizing other tests?

Line 631: therefore repeated

Author Response

Thank you for your valuable comments and suggestions. Based on your comments and suggestions, we make the following changes:

response:

We have made detailed corrections for the spelling, grammar and other errors in the manuscript

Line 59 and 79: Gene types has been modified to genotypes (lines 65)

Oriniformes and orniformes mollusc should be Ornithodoros

Line 124-5: ornithodoros soft ticks and other insects have been described clearly (lines 174-176)

Line 49: have been modified to Africa (lines 195)

section 4: the whole paragraph has been modified  (lines 194-295)

ASF, ASFV, CSF, PCR and so on have been distinguished in the entire manuscript

Line 274 – 271: repeated sentences have been deleted (lines 363-366)

Line 296:  has made a better statement (lines 384-385)

Line 299: have been modified to non‐haemadsorbing (lines 386)

Line 315: have been modified to detect ASFV DNA of ticks  (lines 404)

tables summarizing other tests have been made 

Line 631:  repeated therefore has been deleted (lines 725 )

Round 2

Reviewer 3 Report

The authors have overall improved the manuscript with their revisions; however, it still suffers from some poor English.  It would be beneficial to have someone fluent in English and with infectious biology backqround to edit before publication.

specific comments:

Couldn’t find any reference to Figure 1 in the text.  Also, dates or year should be added to Figure 1.

Lines 145-6, 157-8: italicize species names.

Line 147: reservoir would be more appropriate than “store”

Lines 151-2: sucking blood from infected warthogs

Lines 152-3: is the virus latent in the adult warthog?

Line 155-6: can’t understand sentence

Line 158: This tick is spread??

Line 168-9: maybe important to point out that it delays but doesn’t prevent.

Line 186: what further studies needed?

Line 217:  “cause asf to lose control” needs to be corrected for English

Line 351: derived maybe more appropriate than separated

Line 357: only viral DNA is synthesized rather than host DNA? Doesn’t make sense

Author Response

Dear reviewers:

Thank you for your second valuable comments and suggestions. Based on your comments and suggestions, we make the following changes:  

response:

A reference to Figure 1 is inserted in the text. The year data is placed in the form of annotations in Figure 1 (lines 142-143, 149-150)

Lines 145-6, 157-8: The species name has been italicized (lines 156-157, 179)

Line 147: store has been modified to: reservoir (line 158)

Lines 151-2: has been modified to: the ticks take a blood meal from the infected juvenile warthogs (line 163)

Lines 152-3: the virus latent in the adult warthog has been modified to: Adult warthogs become virus carriers after being infected with ASFV (line 164)

Line 155-6: the sentence has been modified to: There is few reports about the sylvatic cycle in West Africa. (line 169)

Line 158: This tick spread has been modified to: This tick spread ASFV by sucking pig blood (line 172)

Line 168-9: has pointed out that it delays but doesn’t prevent. (lines 182-183)

Line 186: further studies is described as: The potential for the development of new sylvatic or pig-tick cycles, for example in West Africa, requires further investigation (lines 200-201)

Line 217: cause asf to lose control has been modified to: The main factors leading to the failure to control ASF in Africa.(line 232)

Line 351: separated has been modified to: derived (line 367)

Line 357: only viral DNA is synthesized rather than host DNA has been modified to: Macrophages do not divide in vitro, and during the process of ASFV infection on porcine macrophages, viral DNA is mainly synthesized (lines 373-374)

In addition, some adjustments have been made to the English grammar and expressions of the entire manuscript.

Thank you again for your suggestions.